# Local Tree Diversity Suppresses Foliar Fungal Infestation and Decreases Morphological but Not Molecular Richness in a Young Subtropical Forest

**DOI:** 10.3390/jof7030173

**Published:** 2021-02-27

**Authors:** Mariem Saadani, Lydia Hönig, Steffen Bien, Michael Koehler, Gemma Rutten, Tesfaye Wubet, Uwe Braun, Helge Bruelheide

**Affiliations:** 1Institute of Biology/Geobotany and Botanical Garden, Martin Luther University Halle-Wittenberg, Am Kirchtor 1, 06108 Halle, Germany; lydia.hantsch@gmx.de (L.H.); steffenbien@hotmail.com (S.B.); m.koehler.1993@web.de (M.K.); ggpmrutten@gmail.com (G.R.); uwe.braun@botanik.uni-halle.de (U.B.); helge.bruelheide@botanik.uni-halle.de (H.B.); 2German Centre for Integrative Biodiversity Research (iDiv) Halle-Jena-Leipzig, Puschstr. 4, 04103 Leipzig, Germany; tesfaye.wubet@ufz.de; 3Senckenberg Museum of Natural History Görlitz, PF 300 154, 02806 Görlitz, Germany; 4Department of Community Ecology, UFZ-Helmholtz Centre for Environmental Research, Theodor-Lieser-Str. 4, D-06120 Halle, Germany

**Keywords:** leaf fungal pathogens, BEF-China experiment, diversity, visual assessment, high-throughput DNA sequencing

## Abstract

Leaf fungal pathogens alter their host species’ performance and, thus, changes in fungal species composition can translate into effects at the tree community scale. Conversely, the functional diversity of tree species in a host tree’s local neighbourhood can affect the host’s foliar fungal infestation. Therefore, understanding the factors that affect fungal infestations is important to advance our understanding of biodiversity-ecosystem functioning (BEF) relationships. Here we make use of the largest BEF tree experiment worldwide, the BEF-China experiment, where we selected tree host species with different neighbour species. Identifying fungal taxa by microscopy and by high-throughput DNA sequencing techniques based on the internal transcribed spacer (ITS) rDNA region, we analysed the fungal richness and infestation rates of our target trees as a function of local species richness. Based on the visual microscopic assessment, we found that a higher tree diversity reduced fungal richness and host-specific fungal infestation in the host’s local neighbourhood, while molecular fungal richness was unaffected. This diversity effect was mainly explained by the decrease in host proportion. Thus, the dilution of host species in the local neighbourhood was the primary mechanism in reducing the fungal disease severity. Overall, our study suggests that diverse forests will suffer less from foliar fungal diseases compared to those with lower diversity.

## 1. Introduction

Pathogenic fungi play a key role in forest ecosystems, as these microorganisms are able to weaken their plant hosts performance, alter the host community structure, and affect ecosystem functions over short and long periods [1,2,3,4]. Such negative effects have also been shown in forestry, where fungal pathogen infestation is a cause of considerable economic damage [5,6,7]. While the effects exerted by pathogens on host communities have been studied intensively [8], it remains less clear how host species richness affects fungal richness and infestation.

Many studies have described negative relationships between host diversity and overall infection rate, infestation or disease spread in grasslands [9,10,11,12,13,14] and forest communities [15,16,17,18,19,20]. There are several mechanisms by which host species richness can result in such effects [15,16,17,18,19]. One of them is the dilution of competent hosts in a community, resulting in a reduced host density [21,22,23,24]. As susceptible tree individuals are diluted in high-diversity communities by non-host tree individuals, disease-transmission is strongly decreased [25].

A precondition of dilution effects is a certain degree of host specificity of the pathogen [24,26,27]. This might vary from highly host-specific fungi, that grow only on a single host species, to fungi that colonize a few host species, to generalist fungi that occur on a wide variety of host species, under certain environmental conditions and phylogenetic combinations [26,28]. As a multiple-host fungus might manifest several lifestyles [29], their host specificity might differ among life stages. In addition, different life stages might have different impacts on the host [30], depending on the host species [26,31].

In the case of high host specificity, host richness effects would be predominantly brought about by dilution effects. Moreover, fungi with a narrow host range can be affected by the proportion of potential host trees in the local neighbourhood, for example, by altered microclimate or transmission over short distances. The direct interaction or close distances between host and non-host trees might increase or decrease the transmission rate. More spatially concentrated host trees facilitate pathogen dispersal [18]. Thus, the host conspecific density or frequency might determine fungal infestation level [32]. Therefore, the proportion of conspecific hosts in the neighbourhood could be an even stronger predictor for pathogen load and richness than host richness per se. Accordingly, several grassland studies have shown that the negative relationship between plant species richness and foliar fungal infestation were related to decreased host proportion in grasslands [33,34] and in croplands [35,36].

In addition to host dilution, several other mechanisms can reduce disease spread in mixed host communities. For example, fungal pathogen infestation can depend on the presence and density of non-host species, which may modify the microclimatic conditions favourably or unfavourably for pathogen growth [16,37]. When microclimatic conditions in high diversity mixtures impair conditions for fungal development, the degree of infestation of a particular fungal pathogen might be reduced as well [38,39]. High humidity or precipitation [40,41] and warm temperatures [42] have been described to promote pathogen growth and development [43,44,45]. Hence, particular tree species in the neighbourhood of a host individual, which alter humidity or temperature, could have effects on the pathogen growth on that host tree. The biotic neighbourhood might strengthen or weaken the host individual, which might also have a potential influence on the host tree’s defense mechanisms against fungal infestations [33,43,44]. Moreover, plants under certain abiotic stresses perform worse and are more susceptible for diseases [46,47]. Accordingly, high-temperature stress [48,49] or drought stress [50,51] have been reported to increase the susceptibility of cultivated and non-cultivated [52] plants to fungal diseases. Another non-dilution mechanism of host diversity can be the barrier role of non-host species against dispersal and deposition of fungal pathogens. This barrier blocks the transmission pathway between susceptible host trees in short distance transmission events such as transmission via air currents [17,21,39].

Foliar fungal community composition and infestation can be studied by several examination methods. Visual examination of their fruit bodies and characteristic symptoms can reveal the identity of the fungal species to the genus or species level, and help in the estimation of their infestation based on the amount of the visual damage [18,19]. However, this traditional method is highly dependent on the observer knowledge and capacity. In addition, often true pathogens infecting leaves are covered by visual inspection while true endophytes might not cause any visible symptoms [53,54,55]. Even so, endophyte infection induces changes in their host’s metabolism, which might subsequently alter their development [56]. In contrast to fungal pathogens [30,31,57], endophytes might be less host-specific and mainly generalists [58,59,60]. The detection of endophytes might require metabarcoding approaches, making use of next-generation sequencing (NGS), if laborious isolation procedures are inapplicable [19,58,61,62,63,64,65].

In most studies, culture-independent methods such as NGS (e.g., [14,27]) or macroscopic and microscopic visual assessment of damaged plant tissue (e.g., [18,22]) were done independently of each other. The main difference between these two techniques is the amount of the fungal taxa identified. We would expect that the foliar fungal community richness detected would be significantly higher applying NGS rather than macroscopic visual assessment [19,66]. Indeed, Nguyen et al. [19] found that the diversity of the foliar fungal community of silver birches (*Betula pendula*) was 11 times higher when detected by NGS than by visual screening. Theoretically, morphologically identified fungal species should represent a subset of the molecularly revealed species. However, this does not have to be the case, if for some reasons the DNA of some fungal species is less well extracted and amplified [19,67]. Several studies have carried out a direct comparison between culture-dependent and culture-independent approaches based on traditional or molecular examinations [67,68,69,70,71,72,73]. So far, few studies have attempted to compare microscopic assessment with meta-barcoding approaches [19,62,64,66].

The objective of our study was to assess the effects of tree species richness on foliar fungal species richness and fungal infestation, using both visual assessment and high-throughput sequencing methods. We made use of the largest biodiversity ecosystem functioning (BEF) tree experiment worldwide, the BEF-China experiment in Jiangxi province [74,75], where we selected tree host individuals differing in the composition of neighbour tree species. We studied the foliar fungal communities both by visual macroscopic assessment covering all identifiable taxa and DNA meta-barcoding using high-throughput sequencing of the fungal ITS rDNA. We hypothesised that (**H1**) tree diversity in the local neighbourhood decreases fungal richness and infestation by either of the two identification approaches, and that (**H2a**) these diversity effects are mainly explained by a high host-specialisation of the fungal species, resulting in (**H2b**) strong negative effects of the proportion of conspecifics in the local neighbourhood. Finally, we tested the hypothesis that (**H3**) there is a strong overlap between the fungal taxa revealed by DNA high-throughput sequencing and by the morphological assessment, which should also result in a correlation of richness measures based on the two approaches.

## 2. Materials and Methods

### 2.1. Study Location and Leaf Sampling

The sampling was carried out in September 2013 in an experimental forest in subtropical SE China (Xingangshan, Dexing city, Jiangxi Province). The BEF-China experiment was established in 2009 (Site A)/2010 (Site B) with a species pool of 40 native tree species. Different extinction scenarios were simulated on a total of 566 plots at the two mountainous sites with tree richness levels of 1, 2, 4, 8, 16 and 24 tree species [75]. Each plot has a plot size of 25.8 × 25.8 m and 400 tree individuals were regularly planted in 20 rows and 20 columns with a planting distance of 1.29 m (horizontal projection) to the next neighbour tree individual. Here, we focus on half of the random extinction scenario at site A. Site A is characterised by an elevation of about 100 to 290 m a.s.l., a mean slope of 27.5°, a mean annual temperature of 15.1 °C, and mean annual precipitation of 1964 mm [76]. The selected host species pool included one evergreen species (*Castanopsis sclerophylla*) with a mean height at the time of sampling of 1.1 m and seven deciduous species (*Castanea henryi*, *Choerospondias axillaris*, *Liquidambar formosana*, *Nyssa sinensis*, *Quercus serrata*, *Sapindus saponaria*, *Triadica sebifera*) with a mean height of 3.3, 4.7, 3.3, 3.9, 1.2, 2.3 and 2.4 m, respectively. The random extinction scenario was based on a broken-stick design, where all eight tree species occurred in the 8-species mixture, which subsequently was divided into two 4-species mixtures with non-overlapping tree species. Both 4-species mixtures were then divided into each two non-overlapping 2-species mixtures, and the four 2-species mixtures were then divided into eight monocultures. In total, this set comprised 15 plots. We used equal sampling intensity per species and richness level, and sampled 10 individuals per tree species and plot, distributed randomly in the inner 12 rows by 12 columns and sampled 10 leaves per individual tree (*n* = 320), distributed randomly over the tree’s crown for visual inspection. A set of random samples corresponding to six tree species (*n* = 175) (*C. henryi, C. sclerophylla, C. axillaris, N. sinensis, S. saponaria, T. sebifera*) and covering the different tree species richness levels were used for high-throughput sequencing. The leaves were dried for 3 days at 60 °C to exclude subsequent infestation by mold. We decided to use dried leaves to provide enough time for fungal identification and estimation of the degree of infestation of each fungus per leaf. All samples were stored separately in dark and dry places, which enables additional and reliable molecular identification five years later. For this, we cut out five discs at random locations from each non-surface-sterilized leaf for DNA extraction, using a cork borer (6 mm of diameter, yielding approximately 7 mg leaf powder).

### 2.2. Foliar Fungal Screening and Species Identification

The upper and lower surface of each leaf was macroscopically screened for fungal symptoms such as necrosis, mycelium, or fruiting bodies. Using standardised light microscopic techniques and appropriate taxonomic literature, fungal species identification was possible to the species or at least to the genus level for approximately two-thirds of the fungi found on the leaves [77,78,79,80,81,82,83]. Two new fungal species were found and described (*Periconiella liquidambaricola* U. Braun, S. Bien and Hönig on *Liquidambar formosana*, *Tubakia chinensis* U. Braun, S. Bien and Hantsch on *Castanea henryi*; [84,85], two fungal species were found on a so far unknown host (*Erysiphe castaneigena* and *Monochaeatia dimorphospora* on *C. sclerophylla*), and the fungus *Pseudocercospora chibaensis*, hitherto only known from Japan, was for the first time detected in China (Appendix A). Some of the fungi identified at the genus level (e.g., *Microthyrium* sp.) were microscopically identical even when occurring on different host species. However, as it is rather speculative whether they belonged to the same fungal species and as we did not find information on their host ranges, we treated them as different fungal species. If more than one fungal species was present in a particular symptomatic necrosis syndrome, they were treated as a fungal complex. We recognised two different fungal complexes, including three and four fungal species, respectively, on *C. henryi*, and one fungal complex with five fungal species on *C. sclerophylla* (Appendix A). Each complex was treated as a single taxon in all subsequent analyses. The damaged area of each identified fungus/fungal complex was estimated per leaf according to eight damage categories ranging from 0 to 0.1, 0.1 to 1, 1 to 5, 5 to 10, 10 to 25, 25 to 50, 50 to 75, and 75 to 100%. These classes were transformed to mean cover values per leaf for subsequent analyses of fungal infestation (0.05, 0.55, 3, 7.5, 17.5, 37.5, 62.5, and 87.5%, respectively).

### 2.3. DNA High-Throughput Sequencing

We extracted genomic DNA from leaf material using (Charge-switch DNA Plant kit (Invitrogen, Waltham, MA, USA)), with slight changes in the original protocol. After adding 40 μL of elution buffer, the clear DNA solution was transferred to 0.2 mL reaction tube and was stored at 4 °C. The DNA quantification was estimated spectrophotometrically using NanoDrop spectrophotometer (NanoDrop 1000 (Thermo scientific, Waltham, MA, USA)). The nuclear ribosomal internal transcribed spacer (ITS) region was used as the universal barcode for fungal species identification. Polymerase chain reaction (PCR) was performed using for amplicon sequencing of fungal communities ITS rDNA gene region with illumina attached tags. The PCR product consisted of 0.8 ng genomic DNA, 1 μL of 10μM of each of the forward ITS1F primer and the reverse ITS4 primer (Metabion, Munchen, Germany), 12.5 μL of KAPA HiFi HotStart ReadyMix (Roche, Bazel, Switzerland) and nuclease-Free water to obtain 25 μL final volume per reaction. A negative control PCR, to which no DNA template was added, was used to detect any sort of possible contamination during preparation of samples. PCR thermocycling was performed using touchdown PCR conditions involving two separate phases after initial denaturation for 5 min at 95 °C: (i) 15 cycles of a hot start for 20 s at 98 °C, annealing for 45 s at 60 °C–50 °C (−1 °C per cycle) and 72 °C for 2 min; and (ii) 25 cycles of 98 °C for 20 s, 56 °C for 45 s and 72 °C for 2 min with a final extension step of 10 min. The PCR was run on an Eppendorf Mastercycler 5345. Amplicons were visualised by gel electrophoresis on 0.5% agarose gels and were purified using the ExoSAP—Clean Up protocol (New England BioLabs^®^ Inc., Ipswich, MA, USA). The amplification resulted in amplicons with a mean length of 750 bp. A second PCR was conducted to receive a unique tag-index-combination using (indices i5/i7). The second PCR product was assembled on 25 μL final volume consisted of 5 μL purified amplicon, 0.5 μL of 10μM of each of the forward and reverse index primers, 12.5 μL of KAPA HiFi HotStart ReadyMix (Roche, Bazel, Switzerland) and 6.5 μL of Nuclease-Free water. The second PCR thermal protocol consisted of an initial denaturation for 3 min at 95 °C, 5 cycles of 98 °C at 20 s, 57 °C at 1 min and 72 °C for 90 s, and a final extension step for 10 min. The PCR products were purified again with ExoSAP. The purified amplicons were pooled in equimolar ratio to produce one fungal ITS rDNA amplicon library. The library was purified using CleanPCR-kit reagent (CleanDTR (CleanNA, Waddinxveen, The Netherlands)). The purified library was used to perform a paired-end sequencing (2 × 300) by the genetics department of the Ludwig Maximilian University of Munich (LMU) using an Illumina^®^ MiSeq Sequencer (Illumina Inc. San Diego, CA, USA).

### 2.4. Bioinformatics Analysis

The bioinformatics analysis was carried out on a Linux high-performance cluster system. The raw sequence data was demultiplexed based on second PCR adapters, we received 32 sequence (fastq) files. The libraries were separated based on first PCR tags into samples using Cutadapt tool [86]. In addition, we excluded sequences without primer sequences and the primer sequences were trimmed later from the samples reads. Extraneous sequences were removed. Reads with fewer than 35% of base pairs with a quality score equal to or greater than 19 were removed, using the fastq_quality_filter from the fastx_toolkit package (Hannon Lab, Cambridge, UK). These quality-filtered reads were further denoised using the cluster unoise and chimeric sequences were removed using the uchime3 denovo implementations of VSEARCH (version 2.2.0) [87]. The sequence reads resulting from the quality and size filtering were clustered into unique operational taxonomic units (OTUs, Chicago, IL, USA) using linkage-based clustering based on 97% similarity using the VSEARCH (version 2.2.0) clustering tools. The filtered sequence reads per sample were matched to our OTUs with setting a 3% similarity cut-off, thus ignoring reads with major errors. Taxon names were assigned to the OTUs by BLAST (blastn) against GenBank nucleotide database (version 2018-09-07). In addition, we used the software MEGAN (version 6.18.11) to parse the BLAST results. MEGAN blast was based on the lowest common ancestor algorithm with 97% similarity, and a 10% bit-score filter. We took out all non-fungal OTUs during this step (six OTUs represent plant DNA and further 121 with unspecified assignment). For clusters containing more than one sequence, a consensus sequence was produced. All previous steps commands are described in the Appendix A. We used the annotation tool FUNGuild to assign the ecological guild of blast results based either on the genus or species level [88].

### 2.5. Data Analysis

The visual macroscopic assessment data were analysed separately from the sequencing data. At the tree individual level, mean fungal richness per tree individual was calculated by summing up the number of fungal species/fungal complexes per leaf and averaging them across all leaves of a selected tree individual. Mean fungal infestation [% leaf area] per individual was derived by summing up the damaged area of all fungal species/fungal complexes per leaf and averaging them across all screened leaves per tree. The OTU richness was counted for all fungal OTUs and separately for those described as mainly plant pathogens.

The differences between the host trees in morphological fungal richness, fungal infestation and OTU richness were analysed using linear mixed effects models with neighbourhood composition as a random factor, followed by analysis of variance (ANOVA) and a Tukey post-hoc test, using the lmerTest package [89]. We calculated the Shannon diversity of the local neighbourhood, using the relative frequency of the different tree species within the eight (or fewer in case of mortality) neighbour trees. The effect of tree richness in the local neighbourhood on morphological fungal richness, fungal infestation and OTU richness (**H1**) was tested with mixed linear effects models with Shannon diversity, tree species and their interaction as fixed factors and neighbourhood composition as a random factor, followed by an ANOVA. We used the lstrends procedure from the lsmeans package to estimate unbiased marginal means (EMMs) and confidence intervals for the slopes of every single species and considered those slopes significant whose confidence intervals did not overlap with zero. We obtained the overall regression across all species by calculating a mixed model with Shannon diversity as the only fixed predictor and adding species identity as an orthogonal random factor in addition to neighbourhood composition. A network analysis was carried out to investigate the host specificity of the identified fungal taxa/complexes per visual assessment and of the detected OTUs (**H2a**), using the bipartite package [90]. In particular, we calculated generality as the mean effective number of pathogen species per host species. To test the effect of the amount of conspecific target tree in the local neighbourhood, we calculated the proportion of conspecific tree species based on the relative frequency of the eight direct neighbour trees (**H2b**). We run the same mixed effects model as above but used the proportion of conspecific trees instead of Shannon diversity as a predictor. The models with the two different predictors (Shannon diversity and proportion of conspecifics) were compared, using models that were calculated with maximum likelihood (ML) instead of restricted maximum likelihood (REML) and comparing Akaike’s information criterion (AIC). To test the effects of non-target species proportion (excluding the target species) on the overall fungal richness, fungal infestation and OTU richness, we ran mixed effects models with tree species and neighbourhood composition as random factors. All calculations were done with R version 4.0.2 (R Core team 2020).

## 3. Results

### 3.1. Fungal Community Composition and Species Specialisation

Using macroscopic and microscopic visual assessment, 31 fungal taxa/fungal complexes were identified on the phyllosphere of all tree species. We found three fungal taxa/fungal complexes on *Choerospondias axillaris*, *Nyssa sinensis*, *Sapindus saponaria* and *Triadica sebifera*, four on *Castanea henryi* and *Liquidambar formosana* and five on *Castanopsis sclerophylla* and *Quercus serrata* (Appendix A). Mean fungal richness per leaf varied significantly among tree species and was significantly higher for trees of *C. henryi, L. formosana* and *Q. serrata* than for individuals of *N. sinensis, T. sebifera, C. sclerophylla* and *C. axillaris* (Figure 1a, Appendix A). Mean fungal infestation was significantly higher in *C. henryi, L. formosana, N. sinensis* and *Q. serrata* than for the other four host species (Figure 1b, Appendix A).

Foliar fungal community composition revealed by DNA high-throughput sequencing of the fungal ITS region from six tree species *C. axillaris*, *C. henryi, C. sclerophylla, N. sinensis, S. saponaria* and *T. sebifera*, resulted in 3,607,196 high-quality sequence reads clustered in 359 operational taxonomic units (OTUs). The number of reads per sample ranged from 10 to 94,747 reads. In total, 232 OTUs belonged to the fungal kingdom (Appendix A), including 154 Ascomycota, 63 Basidiomycota and 15 fungal taxa remained unassigned. The most common and abundant OTU was assigned to *Cladosporium* sp. (OTU1) (Appendix A) which was present in 90.8% of the samples. According to FUNGuild, 65 OTUs were described as only plant pathogens and they presented 27.4% of the found fungal community (Appendix A, Appendix A). OTU richness per leaf varied from one to 90 OTUs, without significant differences among tree species (Figure 1c, Appendix A).

### 3.2. Tree Species Diversity of the Local Neighbourhood Effect

Based on visually assessed fungal species, we detected an overall marginally significant negative effect of Shannon tree diversity of the local neighbourhood on fungal richness per leaf (Figure 2a, Table 1). However, the response differed among tree species. While the fungal richness of *C. henryi* and *S. saponaria* decreased, fungal richness of *C. axillaris* increased (Table 2). Similarly, fungal infestation per leaf decreased with Shannon diversity of the local neighbourhood (Figure 2b, Table 1). This negative relationship was mainly driven by *C. henryi, L. formosana, S. saponaria, N. sinensis* and *T. sebifera* (Table 2). In contrast to the morphology-based analysis, fungal OTU richness was only marginally significantly related to Shannon diversity of the local neighbourhood (Figure 2c, Table 1). The pathogen community mainly drove this negative slope pattern (Appendix A, Table 1). Here, *T. sebifera* was the only species with a strongly significant negative slope (Table 2).

### 3.3. Effects of the Proportion of Target and Non-Target Tree Species in the Local Neighbourhood

Using the proportion of the host target trees in the local neighborhood as a predictor for morphological fungal richness and fungal infestation resulted in opposite patterns to those of Shannon diversity (Figure 3a,b). In particular, an increasing proportion of conspecifics increased fungal richness of *C. henryi* and *S. saponaria*, while the fungal richness of *C. axillaris* decreased (Table 2). Similarly, the fungal infestation of *C. henryi* and *L. formosana* increased with the proportion of conspecifics in the local neighbourhood (Table 2). The AIC of the models using the proportion of conspecifics in the local neighbourhood was lower than of those using Shannon diversity as a predictor for fungal richness (AIC 368.6 vs. 376.7) and fungal infestation (AIC 1934.0 vs. 1943.4), respectively (Table 1). Consistently, OTU richness was unaffected by the proportion of conspecifics in the local neighbourhood (Figure 3c, Table 1). For OTU richness, the model with the proportion of conspecifics showed an AIC higher (1487.7) compared to Shannon diversity (1483.5). The variation in non-target species proportion did not show any significant effect on fungal richness and OTU richness of all the other non-target species (seven each, Appendix A). However, a higher proportion of *L. formosana* in the local neighbourhood showed a significantly negative effect on the fungal infestation of all the seven other host-species (Appendix A, Appendix A).

### 3.4. Comparison of Foliar Fungal Community Composition Identification Methods

Across the studied host species within the BEF-China experiment, all fungal taxa identified to species level and all fungal complexes both recognised by visual assessment were found on particular host species. Accordingly, the network analysis revealed a generality coefficient of 1.0 host species per pathogen species. In contrast, the generality of OTU richness was 4.24, thus revealing a much higher generality for taxa identified by metabarcording.

Fungal richness assessed visually and fungal richness obtained from high-throughput sequencing were not related (*p* = 0.361, Appendix A). There was only one fungal taxon at the species level (*Colletotrichum gloeosporioides*) that was detected both by visual assessment and high-throughput sequencing. A further six taxa were identified by both methods at the genus level (*Phyllosticta* sp., *Pestalotiopsis* sp., *Monochaetia* sp., *Phyllactinia* sp., *Ramichloridium* sp. and *Pseudocercospora* sp.).

## 4. Discussion

Our study revealed clear effects of local tree diversity on foliar fungal richness and infestation on common Chinese tree species. Moreover, we show that the proportion of target trees in the local neighbourhood better explained by fungal richness and infestation than by Shannon diversity, indicating that dilution effects are important for diversity effects. However, the two approaches, visual assessment and DNA high-throughput sequencing, differed in their ability to detect these relationships. While the expected decrease in fungal richness with local diversity was supported by the visual assessment, there was no such relationship with OTU richness derived from the NGS approach. The results from the two approaches also strongly differed in the degree of fungal specialisation and the identity of the taxa identified.

Based on the visually detected fungal community, local tree species diversity showed a negative effect on fungal richness and fungal infestation, thus confirming **H1**. In particular, we found support for a decrease in overall fungal infestation with the increase in local tree diversity, which has been described before in studies on temperate tree species [18] and grasslands [9,91]. As in the study of Hantsch et al. [18], not all host species responded similarly to the tree diversity in the local neighbourhood. These species-specific responses became even more obvious when basing fungal richness on molecular data, thus including all fungal endophytes. While certain tree species showed positive or negative effects on OTU richness, there was no overall clear trend, as shown before [24]. This discrepancy in the different approaches to detect fungal diversity was also described by Nguyen et al. [19] for birch (*Betula pendula*) in a boreal tree diversity experiment. Possible reasons for these findings are discussed below.

The strong dependence of fungal richness and infestation on the proportion of the target tree in the local neighbourhood points to dilution effects as the main underlying mechanism of this diversity effect [32,92]. An important precondition of dilution effects is a certain host-specificity [26]. The observed strong reduction in fungal infestation with tree diversity in the visual assessment has to be interpreted in the knowledge that the visual assessment also indicated exclusive host-specificity, thus confirming **H2a**. As predicted by the dilution hypothesis, the decrease in density of host species in the local neighbourhood resulted in a lower pathogen load and healthier hosts [21,93]. In contrast, as the pathogen specificity based on OTU richness was much lower, no such relationship was found when being based on NGS. However, we cannot exclude that other mechanisms, such as improved tree growth in more species-rich stands [74] or poorer microclimatic conditions in high diversity mixtures [38,39], might have contributed to this result. Our finding that the models for fungal richness and infestation based on the proportion of the target tree in the local neighbourhood were more parsimonious than those based on Shannon diversity fully supports **H2b**, which suggests that host dilution is the prevailing mechanism [32]. Furthermore, we did not detect strong effects of the proportion of the non-target species at the local scale, except for the negative effect exerted by the presence of *Liquidambar formosana*. This would be expected if non-dilution effects were important. This finding contrasts with the results reported by Hantsch et al. [18], who described a strong effect of the proportion of non-host species on fungal richness and infestation of *Tilia cordata* and *Quercus robur*. Possibly, these variations of non-host driven tree diversity effects on the foliar fungal community dependent on the forest site and tree species composition [27].

In contrast to **H3**, we did not encounter a wider overlap of fungal taxa revealed by visual inspection or by DNA high-throughput sequencing. This was evident in the absence of a correlation in fungal richness detected by these two methods, but also in the low number of taxa that were identified by both approaches. There was only one fungal taxon at the species level and six taxa at the genus level that corresponded between both methods. However, assigning taxa to the species level based on NGS fungal sequences data might not be reliable [71]. Furthermore, the fungal species detected by NGS were found more often on more than one particular host in comparison to those detected by visual inspection, which prevented dilution effects as discussed above. Gilbert and Webb [30] suggested that most pathogens in a tropical forest colonise multiple locally available host trees but that most of the hosts were resistant and did not develop symptoms. This would explain why many fungal OTUs were ubiquitous in the NGS approach. Moreover, we cannot exclude that we only detected spores of these fungi on the leaf surface and that the fungi did not actually occur in the leaf itself. However, also studies in which leaf surface sterilisation was applied showed patterns of ubiquitous fungal endophytic taxa across many different hosts [94,95]. In addition, different life stages on different hosts might have contributed to this pattern [29]. For example, the occurrence of the OTU assigned to *Colletotrichum gloeosporioides* in all six analysed tree species compared to only one species (*C. henryi*) when being identified visually, might be caused by the different forms in which the fungus occurs [96,97]. Weir at al. [97] pointed out the limitations of distinguishing this species from others within the *C. gloeosporioides* complex based on the ITS region. Indeed, Jayawardena et al. [71] recommended that the identification based on the ITS marker should be reported on the genus level or even higher taxonomic levels. Then, morphologically detected fungal taxa in our study might not occur in the sequencing data simply because they are phylogenetically closely related species, for which sequence data has not yet been stored in the GenBank nucleotide database [98]. Although the ITS marker should allow assigning the unknown sequences to lower taxonomic levels (genus or species) [99], for some fungal taxa the identification might be less accurate than the microscopy assessment method [100]. In addition, visual detection of hyphae and reproductive structures does make sure that the fungus was actually actively growing on or within the leaf. Although visual inspection and NGS were carried out on the same leaves, not the whole leaves were used for NGS rather than smaller selected sections. In addition, a lower number of leaves were analysed by tree species in the NGS approach. Thus, it could be that a high variation in fungal composition among leaves on the same tree and the absence of the two non-tested tree species has contributed to the discrepancy between both approaches. However, Nguyen et al. [19] did not detect any positive linear relationship between the fungal taxa comparing both approaches even when applied to the same sampled set. Finally, the storage conditions of the leaf material prior to DNA extraction can affect the fungal community composition and dynamics [101,102]. A consequence of the dependence of NGS methods on storage time and conditions is the requirement of either highly standardized sample or DNA storage conditions or of using NGS methods in combination with other assessment methods. In consequence, we concluded that the visual assessment and the high-throughput sequencing are complementary approaches for assessing the fungal community composition [19,62,66].

## 5. Conclusions

We provide the first evidence that tree species diversity in the local neighbourhood reduces fungal infestation and richness in a subtropical forest. Furthermore, we show that these diversity effects are brought about predominantly by dilution effects. We show that both visual assessment and molecular sequencing of foliar fungal communities have advantages and we urge future studies to use both complementary approaches. Our findings suggest that more diverse forests suffer less from foliar fungal diseases, which could be reflected in increased productivity and reduced tree mortality. Thus, our findings provide another important argument for considering tree species richness as an important component in the establishment of new plantations.

## Figures and Tables

**Figure 1 jof-07-00173-f001:**
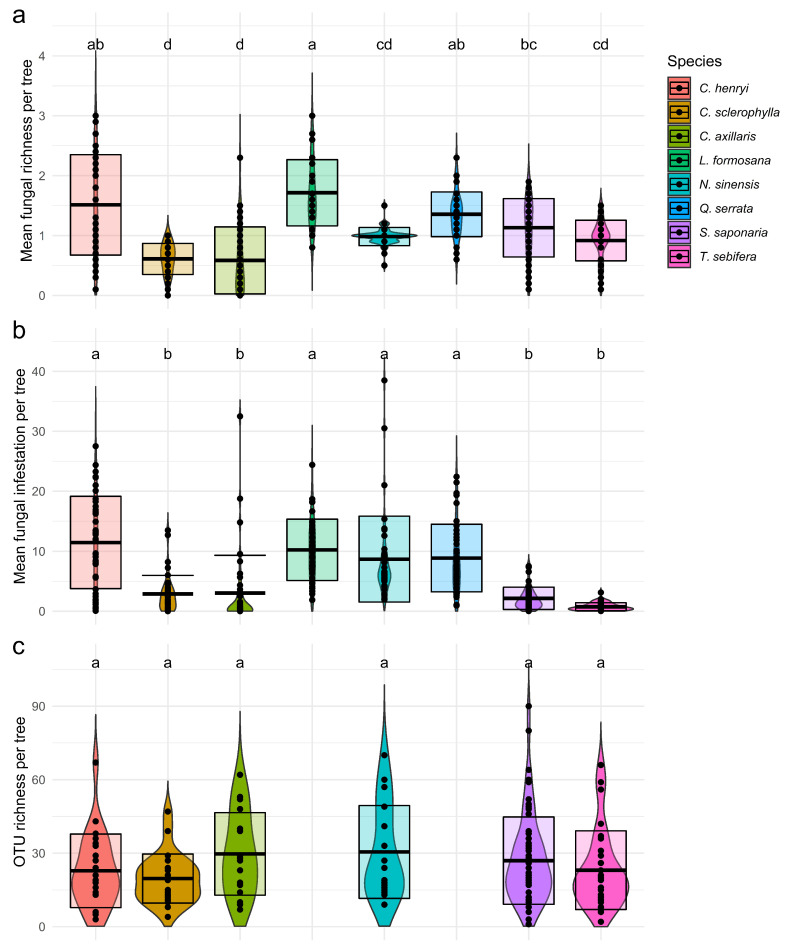
Difference between the eight studied tree species with respect to (**a**) mean fungal richness and (**b**) mean fungal infestation, both based on visual assessment, and (**c**) OTU richness. Letter combinations indicate significant differences between tree species according to a Tukey post-hoc test applied to the linear mixed effects model. (**a**) *p* < 2.2 × 10^−16^, (**b**) *p* < 2.22.2 × 10^−16^, (**c**) *p* = 0.1755.

**Figure 2 jof-07-00173-f002:**
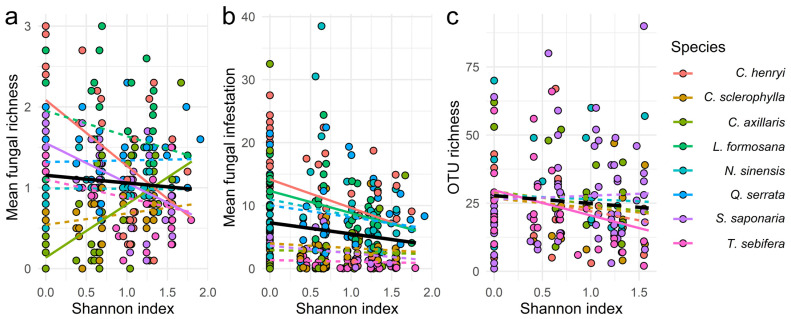
Effects of Shannon diversity in the local neighbourhood on (**a**) mean fungal richness and (**b**) mean fungal infestation, both based on visual assessment data and (**c**) on OTU richness. The overall regression line (in black) was obtained from a mixed model with Shannon diversity as the fixed factor and species identity and neighbourhood composition as random factors (**a**) *p* = 0.0330, (**b**) *p* = 0.0041, (**c**) *p* = 0.0645. The species-specific regression lines (in colour) were obtained from a mixed model with Shannon diversity and species identity as the fixed factors and neighbourhood composition as random factor. Regression solid and dashed lines represent significant (*p* < 0.05) and non-significant (*p* > 0.05) effects, respectively, according to a non-overlap of the confidence intervals of the unbiased predictors for the slopes with zero.

**Figure 3 jof-07-00173-f003:**
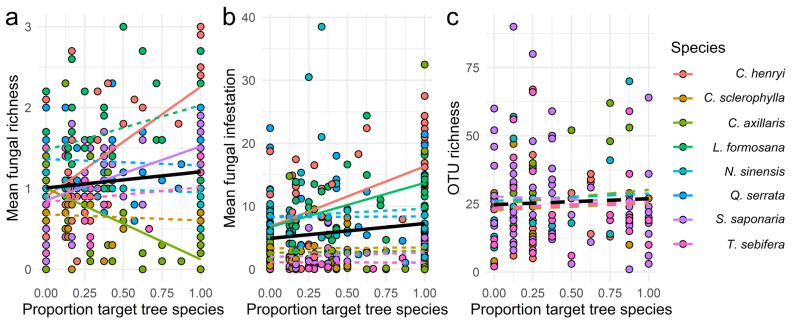
Effects of the host tree species proportion in the local neighbourhood on (**a**) mean fungal richness and (**b**) mean fungal infestation, both based on visual assessment, and (**c)** on OTU richness. The overall regression line (in black) was obtained from a mixed model with Shannon diversity as the fixed factor and species identity and neighbourhood composition as random factors (**a**) *p* = 0.0138, (**b**) *p* = 0.0154, (**c**) *p* = 0.2875. The species-specific regression lines (in colour) were obtained from a mixed model with Shannon diversity and species identity as the fixed factors and neighbourhood composition as random factor. Regression solid and dashed lines represent significant and non-significant effects respectively, according to a non-overlap of the confidence intervals of the unbiased predictors for the slopes with zero.

**Table 1 jof-07-00173-t001:** Analysis of variance (ANOVA) table of the effects of Shannon diversity (Shannon div.) and proportion of the target tree in the local neighbourhood in interaction with species identity (Sp.) on (a) fungal richness, (b) fungal infestation, (c) operational taxonomic unit (OTU) richness and (d) pathogen OTU richness across all tree species. The results are based on linear mixed effects models with neighbourhood composition as random factor. Significant (*p* < 0.05) and marginally significant (*p* < 0.1) *p* values are indicated by bold and italic fonts, respectively.

	Response Variables											
	(a) Fungal Richness			(b) Fungal Infestation			(c) OTU Richness			(d) Pathogen OTU Richness		
Predictor Variables	F Value	*p* Value	AIC	F Value	*p* Value	AIC	F Value	*p* Value	AIC	F Value	*p* Value	AIC
Shannon div.	4.82	**0.0331**	376.7	8.59	**0.0042**	1943.4	3.62	*0.0645*	1483.6	3.98	*0.0517*	1038.4
Sp.	18.35	**<0.0001**		8.81	**<0.0001**		1.43	0.2418		0.74	0.5975	
Sp. × Shannon div.	8.93	**<0.0001**		1.96	*0.0635*		1.84	0.1262		1.45	0.2194	
Proportion of target tree	6.39	**0.0138**	368.6	6.05	**0.0154**	1934	1.16	0.2875	1487.7	2.35	0.1310	1038.9
Sp.	8.07	**<0.0001**		7.64	**<0.0001**		1.16	0.3346		2.68	**0.0264**	
Sp. × Proportion of target tree	10.94	**<0.0001**		3.42	**0.0025**		1.06	0.4023		1.53	0.2005	

**Table 2 jof-07-00173-t002:** Statistical analysis of Shannon diversity (Shannon div.) and proportion of the target tree at the local neighbourhood effects on (a) fungal richness, (b) fungal infestation, (c) OTU richness and (d) pathogen OTU richness per tree species. Linear mixed effects models with Shannon diversity or proportion of target tree in interaction and tree species as predictor variables and neighbourhood composition as a random factor. Significant values are those with confidence interval that do not overlap with zero, indicated by bold fonts.

		Response Variables											
		(a) Mean Fungal Richness			(b) Mean Fungal Infestation			(c) OTU Richness			(d) Pathogen OTU Richness		
Predictor Variables	Tree Species	Slope	Lower.CL	Upper.CL	Slope	Lower.CL	Upper.CL	Slope	Lower.CL	Upper.CL	Slope	Lower.CL	Upper.CL
Shannon div.	*C. henryi*	**−0.8956**	**−1.2200**	**−0.5716**	**−7.0420**	**−10.880**	**−3.2080**	−11.550	−29.740	6.6400	−3.5205	−8.8400	1.8000
	*C. sclerophylla*	0.1232	−0.2050	0.4515	−0.1020	−3.9900	3.7910	5.2100	−9.8200	20.240	0.0964	−4.3700	4.5610
	*C. axillaris*	**0.7420**	**0.4370**	**1.0467**	0.5120	−3.1000	4.1270	−16.630	−36.710	3.4500	−4.7956	−10.810	1.2190
	*L. formosana*	−0.2835	−0.5950	0.0282	**−4.4210**	**−8.1300**	**−0.7150**	_	_	_	_	_	_
	*N. sinensis*	−0.0088	−0.3300	0.3127	−1.9670	−5.7800	1.8490	−2.2600	−18.170	13.650	−0.0387	−4.7000	4.6200
	*Q. serrata*	0.0645	−0.2200	0.3489	0.1520	−3.2100	3.5150	_	_	_	_	_	_
	*S. saponaria*	**−0.5462**	**−0.8550**	**−0.2377**	−2.6540	−6.3000	0.9920	4.0500	−7.0700	15.170	1.0171	−2.3200	4.3520
	*T. sebifera*	−0.2655	−0.5780	0.0475	−0.6080	−4.3300	3.1140	**−16.380**	**−31.590**	**−1.1700**	**−4.6167**	**−9.1100**	**−0.1220**
Proportion of target tree	*C. henryi*	**1.5033**	**1.0258**	**1.9810**	**12.334**	**6.8000**	**17.870**	5.4200	−22.770	33.600	2.3690	−5.4480	10.180
	*C. sclerophylla*	−0.0516	−0.5797	0.4760	−0.4090	−6.5700	5.7500	−1.9700	−29.900	25.970	0.9990	−7.0560	9.0500
	*C. axillaris*	**−0.9588**	**−1.3567**	**−0.5610**	0.3180	−4.3000	4.9400	24.290	**−3.7600**	52.340	**8.1890**	**0.1990**	**16.180**
	*L. formosana*	0.4794	−0.0368	0.9960	**8.6380**	**2.6300**	**14.640**	_	_	_	_	_	_
	*N. sinensis*	−0.1039	−0.6348	0.4270	−1.0230	−7.2200	5.1700	14.330	−19.010	47.680	4.6770	−4.8760	14.230
	*Q. serrata*	−0.1692	−0.6083	0.2700	−2.0440	−7.1400	3.0500	_	_	_	_	_	_
	*S. saponaria*	**0.7642**	**0.2779**	**1.2510**	0.9010	−4.7300	6.5300	−10.060	−28.950	8.8200	−3.8130	−9.2600	1.6300
	*T. sebifera*	0.2105	−0.2979	0.7190	0.2640	−5.6600	6.1800	2.4700	−20.880	25.830	1.6460	−5.0890	8.3800

## Data Availability

The used data in this study is not reported yet. But it is available if requested from the corresponding author.

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
