# Peer review of "Local Tree Diversity Suppresses Foliar Fungal Infestation and Decreases Morphological but Not Molecular Richness in a Young Subtropical Forest"

_jof, 2021, doi:10.3390/jof7030173_

Round 1

Reviewer 1 Report

This is an interesting ecological research on a forest from China. However, there are some points that should be addressed in order to make the data clearer and the results more relevant.

General comments:

  • The introduction is too long and should be shortened. The text should be more concise and assertive.
  • The last paragraph of your introduction should not have so many details. Detailed information should be in the methods, presented in the results and then discussed afterwards.
  • Some sentences are also too long. They will be easier to read and understand if shortened, divided into several sentences, or if punctuation is added. Note that this is an issue throughout the entire manuscript.
  • In the methods section, it is not clear when was the work performed. Since the sampling was done in 2013, was all the other experimental work done at that time? Or, did you use around 7-years old samples? This is very important information that can impact on your results and discussion, and it might even explain some of your results. Please make sure that it is clear how and when did you do all the different steps. If you used old samples, make sure to describe in detail how they were kept and preserved after sampling. Also, if you did the sampling explain how you did it, or if this was done for an already published paper, make sure you explain this in a clear way.
  • In all your figures, make sure the scientific names are written in italics. If possible, increase the resolution for figure 1, and increase the size of figure 2.

Specific comments:

Lines 122 and 128: The sentences within this paragraph should be in the same verbal tense.

Line 128: The acronym BEF should be written in full here, since this is the first mention in the manuscript text (abstract is excluded from this).

Line 169: Instead of “mold fungi”, it should read: “filamentous fungi”, or “mycelial fungi”, or “mold”.

Line 179: When you mention the references [77-83], are those for the methods? Do they describe how to perform the method? Or was the work actually done for other research papers? It is not clear. The same applies to the references in line 181.

Line 199: The DNA extraction was done how long after sampling? How were the leafs kept until being processed for DNA extraction?

Line 219: Did the second PCR followed the exact same parameters as the previous one? And were the amounts of the different constituents the same as described before? If so, please mention it. If not, please described as you did before.

Line 342: There is a typo in the word “proportion” on the last row of the first column in table 1, please revise.

Line 474: Where it reads “we would conclude…”, it should read “we concluded…”.

Line 484: It is not common to cite any other work on the conclusions of a research paper. Unless contextualized, I suggest removing this reference.

Round 2

Reviewer 1 Report

All the previous comments were addressed and all issues were sorted. The current manuscript reads much better and I have no further comments or suggestions.